# Quantitating Wastewater Characteristic Parameters Using Neural Network Regression Modeling on Spectral Reflectance

**Dhan Lord B. Fortela** [1,2,*], **Armani Travis** [2], **Ashley P. Mikolajczyk** [1,2], **Wayne Sharp** [1,3], **Emmanuel Revellame** [1,2], **William Holmes** [1,2], **Rafael Hernandez** [1,2] and **Mark E. Zappi** [1,2]

[1] Energy Institute of Louisiana, University of Louisiana, Lafayette, LA 70504, USA; ashley.mikolajczyk@louisiana.edu (A.P.M.); wayne.sharp@louisiana.edu (W.S.); emmanuel.revellame@louisiana.edu (E.R.); william.holmes@louisiana.edu (W.H.); rafael.hernandez@louisiana.edu (R.H.); mark.zappi@louisiana.edu (M.E.Z.)

[2] Department of Chemical Engineering, University of Louisiana, Lafayette, LA 70504, USA; armani.travis1@louisiana.edu

[3] Department of Civil Engineering, University of Louisiana, Lafayette, LA 70504, USA

[*] Correspondence: dhanlord.fortela@louisiana.edu

**Abstract:** Wastewater (WW) analysis is a critical step in various operations, such as the control of a WW treatment facility, and speeding up the analysis of WW quality can significantly improve such operations. This work demonstrates the capability of neural network (NN) regression models to estimate WW characteristic properties such as biochemical oxygen demand (BOD), chemical oxygen demand (COD), ammonia ($NH_3$-N), total dissolved substances (TDS), total alkalinity (TA), and total hardness (TH) by training on WW spectral reflectance in the visible to near-infrared spectrum (400–2000 nm). The dataset contains samples of spectral reflectance intensity, which were the inputs, and the WW parameter levels (BOD, COD, $NH_3$-N, TDS, TA, and TH), which were the outputs. Various NN model configurations were evaluated in terms of regression model fitness. The mean-absolute-error (MAE) was used as the metric for training and testing the NN models, and the coefficient of determination ($R^2$) between the model predictions and true values was also computed to measure how well the NN models predict the true values. The highest $R^2$ (0.994 for training set and 0.973 for testing set) and lowest MAE (0.573 mg/L BOD, 6.258 mg/L COD, 0.369 mg/L $NH_3$-N, 6.98 mg/L TDS, 2.586 m/L TA, and 0.014 mmol/L TH) were achieved when NN models were configured for single-variable output compared to multiple-variables output. Hyperparameter grid-search and k-fold cross-validation improved the NN model prediction performance. With online spectral measurements, the trained neural network model can provide non-contact and real-time estimation of WW quality at minimum estimation error.

**Keywords:** neural network regression; wastewater quality; spectral reflectance



## 1. Introduction

This study demonstrates for the first time the potential of neural network (NN) regression models [1] in estimating wastewater (WW) quality parameters (BOD, COD, $NH_3$-N, TDS, TA, and TH) by using WW spectral reflectance as the model input. The task of treating WW is integral in the objective of minimizing the negative footprint of human waste in our environment and its impact on human health. Hence, WW treatment facilities exist in various systems involving WW streams, from industrial to municipal sectors [2]. To maintain these facilities within their set continuous operating conditions while meeting regulatory levels of effluent streams, various parameters are monitored [3], such as BOD, COD, $NH_3$-N, TDS, TA, and TH, to check the quality of effluents. A common approach of monitoring is by sampling the WW at various locations of the treatment facility and the samples are tested immediately using readily available test kits or by being brought to the laboratory to undergo various physical and chemical analysis procedures. Some of these

procedures can take hours to days to be completed (COD, BOD, etc.) [2]. The lag time in making operational decisions based on WW quality measurements can significantly affect the dynamics of the treatment facility amid the control systems in place [4–6]. Recent works on the analysis of water quality by [7–9] have shown the potential of visible to near-infrared spectral reflectance data as a reliable spectral signature of key WW parameters (BOD, COD, etc.). The data analytics implemented in these previous works consist of several mathematical transformations that involved human intervention in selecting a set of good wavelength bands for each WW parameter [7,9]. Amid achieving high $R^2$ [7], such a data analysis approach still needed human intervention to complete the regression modeling and prediction. This approach may be fitting when the objective is a highly accurate estimation of water or WW parameters not for use in process control, e.g., for testing river or lake water quality, but it cannot be easily integrated into a real-time estimation for process dynamics control in a WW treatment facility. Hence, a fast method of estimating WW quality parameters would be desirable in achieving the goal of operating a WW treatment facility within target operational settings.

We propose in this work the use of NN regression models to accelerate the estimation of WW quality parameters. A trained NN can be deployed and used in the estimation of WW parameters in the time-scale of milliseconds to minutes [1], which is fitting in a process control of a WW treatment facility. In addition to the theoretical basis of the reliability of spectral reflectance for building estimation models [10], others have demonstrated the potential of artificial NN in using spectral reflectance data to model and estimate water quality parameters in large water bodies [11], properties of soil and rocks [12], and properties of crops [13,14]. The success of these works indicates the possible applicability of NN models in other, similar tasks such as estimating WW properties with potential applications in the process control of WW treatment facilities. Another advantage of developing NN models, given the current developments in computing, is their adaptability for deployment in machine learning-dedicated hardware designed for mobile applications, e.g., NVIDIA Jetson Nano [15]. The main objective of the study is to demonstrate the potential of NN as regression models that can estimate WW characteristic parameters at minimal prediction errors. Here are the specific questions that this paper aims to answer:

- What data preprocessing tasks must be carried out on the WW spectral data to produce regression NN models with good prediction performance?
- What are the effects of common hyperparameters in NN modeling, including the number of hidden layers and number of neuron units in each hidden layer?
- How does the number of modelled outputs, i.e., WW parameters, affect the prediction performance of NN models?
- How can hyperparameter tuning and k-fold cross-validation on regression NN models improve prediction performance?

There are more pertinent questions that may be asked and tested to evaluate the capabilities of NN models for the regression of WW characteristic parameters due to the numerous possible ways of setting the architecture of NN models, and due to the complexities that certain WW datasets can pose. Nonetheless, the questions above set to be answered in this study should elaborate on important aspects of adopting NN models to quantitate WW characteristic parameters by modelling on WW spectral reflectance data. The contribution of this work in the current literature is two-fold: (1) the development of a data analytics workflow to estimate WW quality parameters using spectral reflectance as input data, and (2) the demonstration of the potential to reduce the computational steps and time of using spectral reflectance data of WW to estimate WW quality parameter levels by using NN regression models as prediction models.

## 2. Methodology

A schematic overview of the data analytics workflow implemented is shown in Figure 1. The NN computations were implemented via Python codes using the Keras-TensorFlow modules for NN modeling [16]. The hardware was a laptop computer with an Intel Core i7

(12th Gen) CPU with 2.10 GHz base speed (max speed 5.3 GHz). The Python codes organized in Jupyter Notebook files used in the data analytics workflow have been made available in the online GitHub repository of the paper [17] (URL: https://github.com/dhanfort/WW_Spectra_Nnlearning.git, accessed on 1 August 2023).

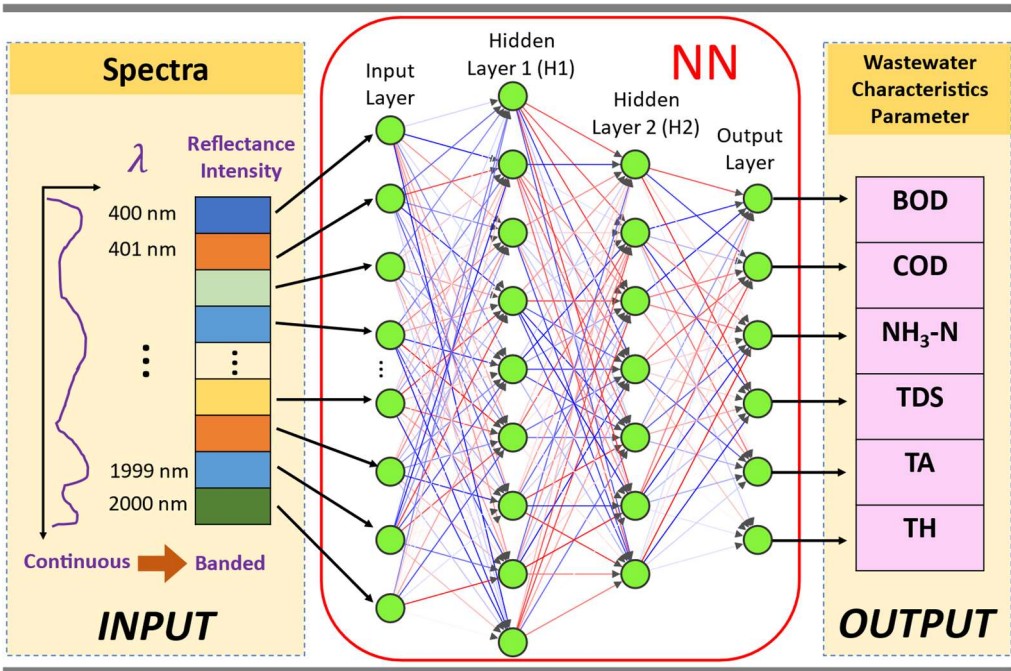

**Figure 1.** Schematic overview of the data analytics workflow implemented in this work. The wastewater spectral reflectance in the wavelength range of 400 nm to 2000 nm raw signal in continuous form is banded into a 1 nm interval. Then, the banded signal is used as input data to train the neural network (NN) model with the levels of wastewater parameters (BOD, COD, NH$_3$-N, TDS, TA, and TH) as target outputs.

*2.1. Dataset*

2.1.1. WW Data Source and Structure Overview

The spectral reflectance dataset used in this study were the open-source data originally collected and published by Xing, Chen [7]. The WW samples were taken from different locations in a municipal WW treatment facility: water inlet (Influent WW), anoxic tank, aerobic tank, sedimentation tank and water outlet (Effluent WW) under different treatment methods at a domestic sewage treatment plant [7]. The various chemical analyses carried out were discussed in detail by Xing, Chen [7]. They noted that there were two subsets of the dataset according to WW quality, consisting of high levels and low levels of COD, BOD, and NH$_3$-N: (Group 1) high levels for influent WW, and (Group 2) low levels for anoxic tank, aerobic tank, sedimentation tank, and water outlet (effluent WW). The collected dataset consisted of WW spectral reflectance in the wavelength of 400 nm to 2000 nm (visible to near infrared) with the corresponding measurements of levels of BOD, COD, NH$_3$-N, TDS, TA, and TH, which were designated as the targets of the learning process. There were a total of 87 data samples in the whole dataset [7]. A descriptive summary of the dataset is shown in Figure 2.

2.1.2. Training Set and Test Set

For each NN modelling, the dataset was divided into two exclusive subsets: the training set and the test set. The training set was used to adjust the parameters of the NN model and in computing the training cost function. The test set was used to compute the cost function MAE on data not seen by the NN Model during training. To keep the

same training and testing datasets across all computational experiments, the random seed index was set to a constant value (see the Python code for the details). When modelling on the whole dataset, subset partitioning was 90% for training and 10% for testing. When modelling on the Influent WW dataset, dataset partitioning was 90% for training and 10% for testing. When modelling on the Group 1 (Influent WW) dataset, the dataset partitioning was 90% for training and 10% for testing. When modelling on the WW Group 2 dataset, the dataset partitioning was 80% for training and 20% for testing. The input data, which had levels of banded spectral intensity (see Figure 1) in both training set and testing set data, were normalized by scaling using the 'MinMaxScaler' of the Scikit-Learn [18]. Scaling input data to a machine learning model being trained has been shown to improve model prediction performance [19]. Note that since the input data was the banded spectral reflectance at 400 nm, 401 nm, . . ., 2000 nm, with each band as a feature, each input sample is a vector of 1601 features.

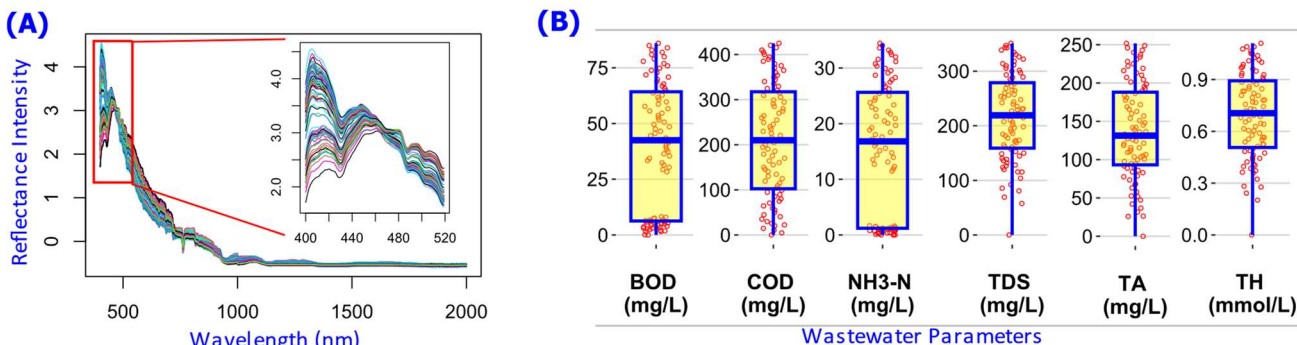

**Figure 2.** Descriptive statistics of the wastewater spectral and quality parameters dataset. (**A**) Plot of all the wastewater samples' reflectance intensity over wavelength (nm), which is the input data for the NN models. (**B**) Boxplots of the levels of wastewater parameters, which are the output data in training the NN models. The dataset is an open-source data originally collected by Xing, Chen [7].

### *2.2. NN Model Training and Testing*

There are various model settings that can be specified when working with NN models. The following subsections describe the main components of the NN modelling implementation. Other details can be found in the Jupyter Notebook files of Python codes for the work [17].

### 2.2.1. Learning Cost Function

The objective of the learning process in NN modelling was to minimize the cost function (also called loss function). Among various possible metrics of machine learning with Keras-TensorFlow [20], the mean-absolute-error (MAE) measures the deviations from the true values more directly [21], which can be clearly interpreted as the average error in estimating the WW parameter levels. $MAE = \sum_i^n \frac{|Y_i - T_i|}{n}$, where $T_i$ = true value of the WW parameters in observation sample $i$ and $Y_i$ = prediction on the levels of the WW parameters of observation sample $i$ by the NN model. The inputs to the NN model $f_{NN}$ were the banded spectral reflectance signal designated as $X_i$; hence, $Y_i = f_{NN}(X_i, \theta)$, where $\theta$ is the NN model parameters configured (hyperparameters) and tuned (neuron weights) during training. Hence, the objective for each NN modelling was to minimize MAE by adjusting $\theta$ via an optimization algorithm. The coefficient of determination ($R^2$) between the model predictions and true values was also computed to measure how well the NN models predict the true values [22].

### 2.2.2. Optimization Algorithm, Learning Rate, Activation Function, and Training Epoch

The Keras-TensorFlow modules for NN modelling contain various optimization algorithms that were all used in this work during preliminary computation runs: Adadelta,

Adafactor, Adagrad, Adam, AdamW, FTRL, Lion, Nadam, RMSprop, and SGD [23]. Among these algorithms, the Adam optimizer consistently produced NN models with the best fit scores (low MAE and high $R^2$). Hence, the Adam optimization algorithm was used in the majority of computations. The best performance of the Adam optimizer was also achieved when the learning rate was set to 0.0001. The training epoch was set to 5000, which was found to be large enough to allow for the cost function MAE to settle at an almost constant level in all of the training runs. The best activation function [24] for the hidden layers (H1 and H2) was also determined to be the 'ReLU' function. The best activation function for the output layer was the 'Linear' function.

2.2.3. Effect of Number of Hidden Layers, Number of Neuron Units, and Outputs

To test the performance of NN models at varying numbers of hidden layers, and the number of neurons in the hidden layers, the following configurations were used: one hidden layer (H1) with low and high numbers of neuron units, and two hidden layers (H1, and H2) with the number of neuron units varied in the first layer (H1) and the number of neuron units fixed in the second layer (H2). See Table 1 for the summary of these hidden layer configurations.

**Table 1.** Summary of NN modelling run settings implemented in the study.

| NN Model Settings | Model Output(s) Y Settings |
|---|---|
| (1) One Hidden Layer: H1 w/ 32 neuron units<br>(2) One Hidden Layer: H1 w/ 1000 neuron units<br>(3) Two Hidden Layers: H1 w/ 64 neuron units;<br>H2 w/ 32 neuron units<br>(4) Two Hidden Layers: H1 w/ 1000 neuron units;<br>H2 w/ 32 neuron units | Multiple Outputs:<br>(1) BOD, COD, NH3-N, TDS, TA, TH (all)<br>(2) BOD, COD, NH3-N, TDS, TA<br>(3) BOD, COD, NH3-N, TDS<br>(4) BOD, COD, NH3-N<br>(5) BOD, COD<br>Single Output:<br>(6) BOD; (7) COD; (8) NH3-N; (9) TDS; (10) TA; (11) TH |
| Total Number of Modelling Settings = (NN Model Settings) × (Model Output(s) Settings) = 4 × 11 = 44 | |

Given that there were several WW parameters being estimated (BOD, COD, NH3-N, TDS, TA, and TH), the effects of various combinations of the WW parameters being modelled as outputs were also evaluated. Table 1 shows a summary of NN modelling implemented consisting of 44 different computational runs. Note that the set of combinations of model output Y (Table 1) does not cover all possible combinations, i.e., there are $2^6 = 64$ possible Model Output(s) Y Settings, resulting in (NN Model Settings) × (Model Output(s) Settings) = 4 × 64 = 256 Total Number of Modelling Settings for all possible settings of computational experiments. Rather, the choice of model output combinations was motivated by the purpose of working with a number of results, i.e., 44 sets of results, that was manageable in the discussion of key aspects of the work.

2.2.4. NN Model Hyperparameter Grid-Search

Often, at the start of developing a NN model and when a very challenging dataset is encountered, the NN model settings for the optimizer, learning rate, number of neurons per hidden layer, etc., must be fine-tuned. This task focuses on finding the NN model parameters other than the weights of neuron connections in the NN Model, and this is called the hyperparameter grid-search, with the main goal of improving the prediction performance of the NN model. The grid-search implemented was the full-factorial search via the Scikit-Learn module 'GridSearchCV' [25], which considers all combinations of hyperparameters being tested. The hyperparameter grid-search was carried out during the preliminary runs to set the optimizer Adam, learning rate 0.0001, epoch of 5000, and activation function 'ReLU' for H1 and H2, as discussed in Section 2.2.2.

The hyperparameter grid-search was also implemented to refine the NN modelling on the Group 2 WW subset that posed a challenge (see Section 3.3). This task demonstrated how

NN models can be subjected to tuning the hyperparameters such as optimizer algorithm, number of neuron units per hidden layer, activation function in each hidden layer and the output layer, and learning rate. The selection of the best NN model with its respective hyperparameter settings was still based on the MAE. The grid-search settings produced a total of 864 NN models evaluated (see Section 3.3).

2.2.5. Repeated K-Fold Cross-Validation during NN Training

When the dataset has a small sample size, the training of the NN model can be improved by performing k-fold cross-validation [1], which exposes the NN model to all data samples one fold subset at a time by dividing the training data into k folds and implementing training k times. Hence, this study also demonstrated the performance of k-fold cross-validation by implementing repeated k-fold training on the Group 2 WW data subset. This was performed using the 'RepeatedKFold' module from Scikit-Learn [26] with k = 3 folds and repeat = 10, where the sampling for each fold at each repeat was randomly initialized to minimize training on the same k-fold subsets for each repeat. A hyperparameter grid-search was also performed and determined the following settings to be the best for the repeated k-fold training: H1 = 1000 units, H2 = 1000 units, and 'Linear' as the best activation function for H1, H2, and output layers. This resulted in training the NN model 30 different times on the Group 2 WW data subset (see Section 3.3).

**3. Results**

Though numerous results were generated in the study, the following results sections have been organized to facilitate the discussion of the key aspects of the work. Section 3.1 covers the results that show the need for dedicated NN models for WW stream groups. Section 3.2 covers the results that show how the performance of NN models can be affected by the various combinations of WW parameters used as model outputs. Section 3.3 covers the results that show how hyperparameter grid-search and k-fold cross-validation can improve the predictive performance of NN models.

*3.1. Need for a Dedicated NN Model for Wastewater Stream Groups*

Part of the initial stage of developing the NN models was the task of determining whether the whole WW dataset could be treated as a single input array to the models. Training the NN model on the whole dataset resulted in a set of predictions that shows two apparent groups of the data, as shown in Figure 3 for the model outputs BOD, COD, and $NH_3$-N. This was an indication that the WW dataset cannot be treated as a single input array. After dividing the whole dataset into two groups according to the Group 1 WW and Group 2 WW subsets, the NN models started to achieve good prediction performance, as shown in Figure 4 for the Group 1 WW (Influent WW) and in Figure 5 for the Group 2 WW. These results indicate the need for dedicated NN models for certain WW stream groups.

*3.2. Challenge with Increasing Number of NN Model Output Variables*

By varying the combination of WW parameters as model outputs (Table 1), the comprehensive evaluation of NN model prediction performance was collected and the key results are shown in Figures 6 and 7. In general, NN models trained for single output, i.e., one WW parameter estimated per NN model trained, achieved the best prediction performance in terms of the $R^2$ and MAE between the prediction Y and actual level T of each WW parameter (Figure 6A–D). When more than one WW parameter is being modeled in an NN model, the $R^2$ and MAE of Y-vs-T may turn to poorer levels (Figure 6A–D), even though increasing the number of hidden layers and the number of neuron units can be used to improve the $R^2$ and MAE of Y-vs-T (Figure 6D). Also, a corresponding decrease in MAE is observed, in general, with an increase in the $R^2$ of Y-vs-T of an NN model. The MAE levels for the best NN models (Figure 6D) can be very low, as shown in Figure 8, which is a good indication of minimizing prediction error.

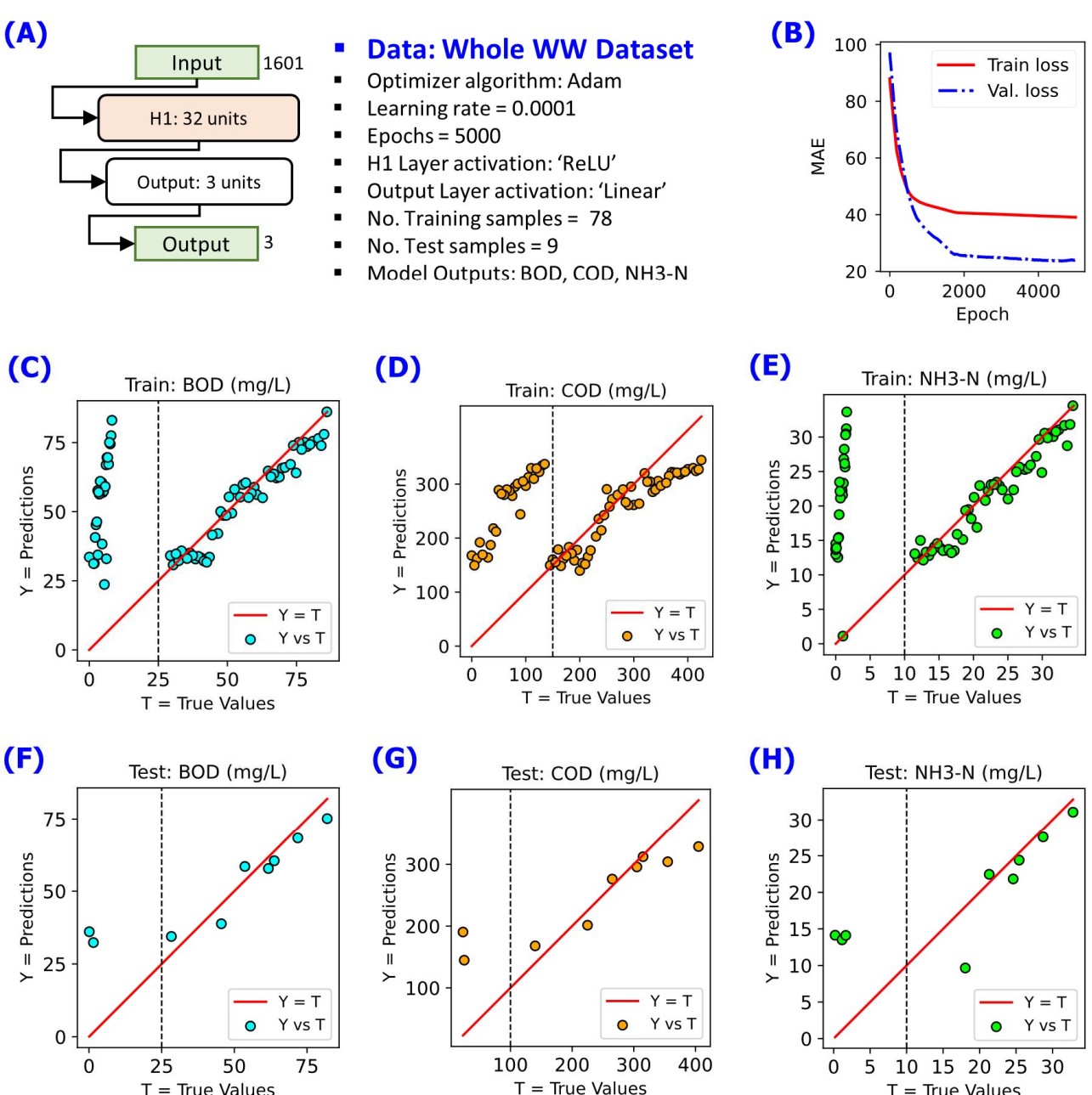

**Figure 3.** Illustration on the need to train an NN model dedicated to a wastewater stream group. One NN model may not be able to capture all trends, as indicated by the two apparent groups in the prediction-versus-actual plots. Results of training, validation, and testing of the NN model with 1 hidden layer of 32 units of neurons. Annotated red solid line shows the reference relation Y = T. (**A**) NN model configuration implemented, (**B**) changes in cost function MAE during the training for 5000 epochs, (**C**) Y-vs-T of the training data for BOD, (**D**) Y-vs-T of the training data for COD, I Y-vs-T of the training data for NH$_3$-N, (**E**) Y-vs-T of the training data for NH$_3$-N, (**F**) Y-vs-T of the test data for BOD, (**G**) Y-vs-T of the test data for COD, and (**H**) Y-vs-T of the test data for NH$_3$-N.

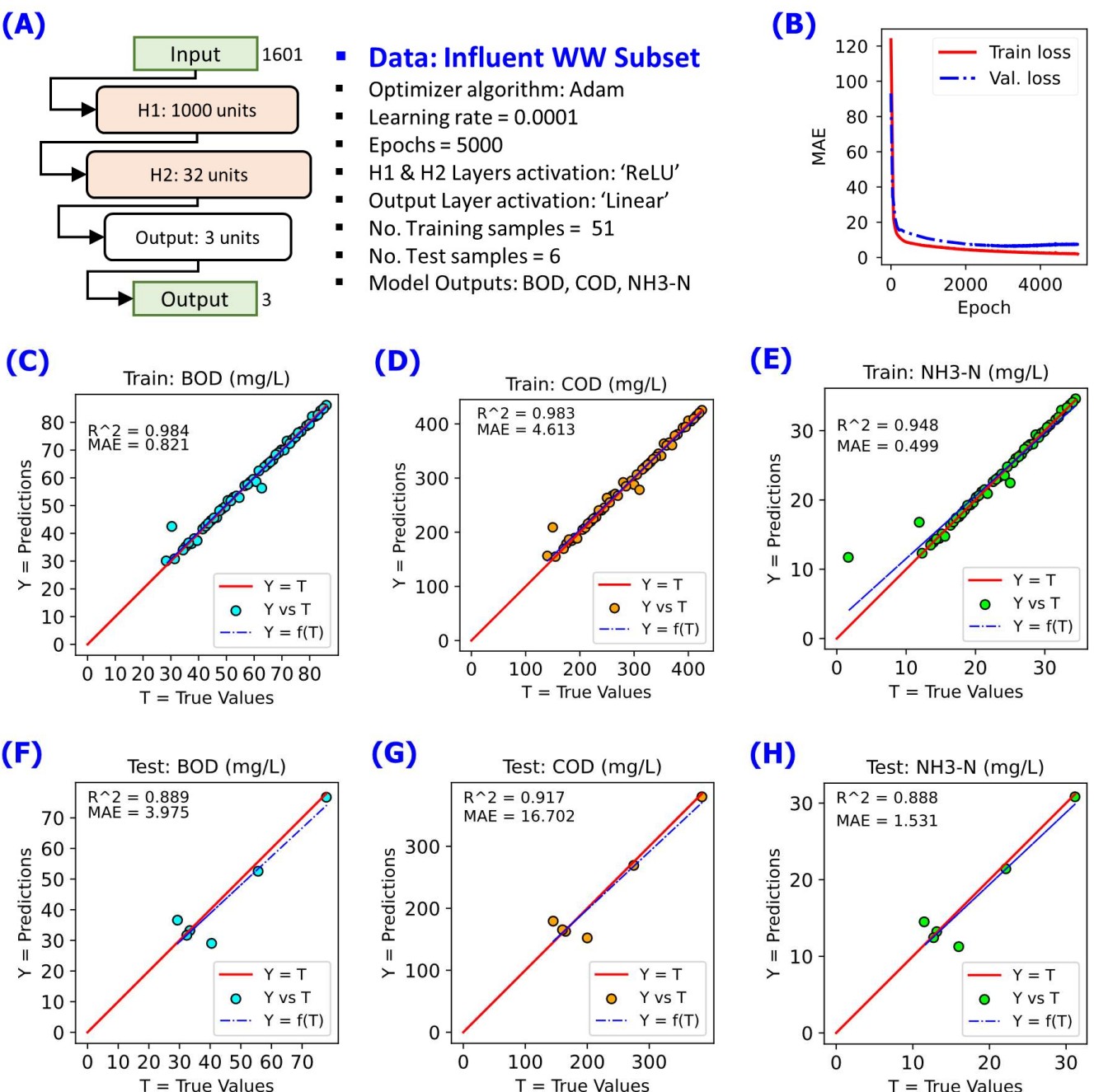

**Figure 4.** NN modelling on the data subset for Group 1 WW (Influent WW) using 2 hidden layers consisting of 1000 neuron units in first hidden layer H1 and 32 neuron units in second hidden layer H2. A linear fit Y = f(T) with a blue dashed line is annotated to compare with reference line Y = T with the red solid line. (**A**) NN model configuration implemented, (**B**) changes in cost function MAE during the training for 5000 epochs, (**C**) Y-vs-T of the training data for BOD, (**D**) Y-vs-T of the training data for COI (**E**) Y-vs-T of the training data for NH$_3$-N, (**F**) Y-vs-T of the test data for BOD, (**G**) Y-vs-T of the test data for COD, and (**H**) Y-vs-T of the test data for NH$_3$-N.

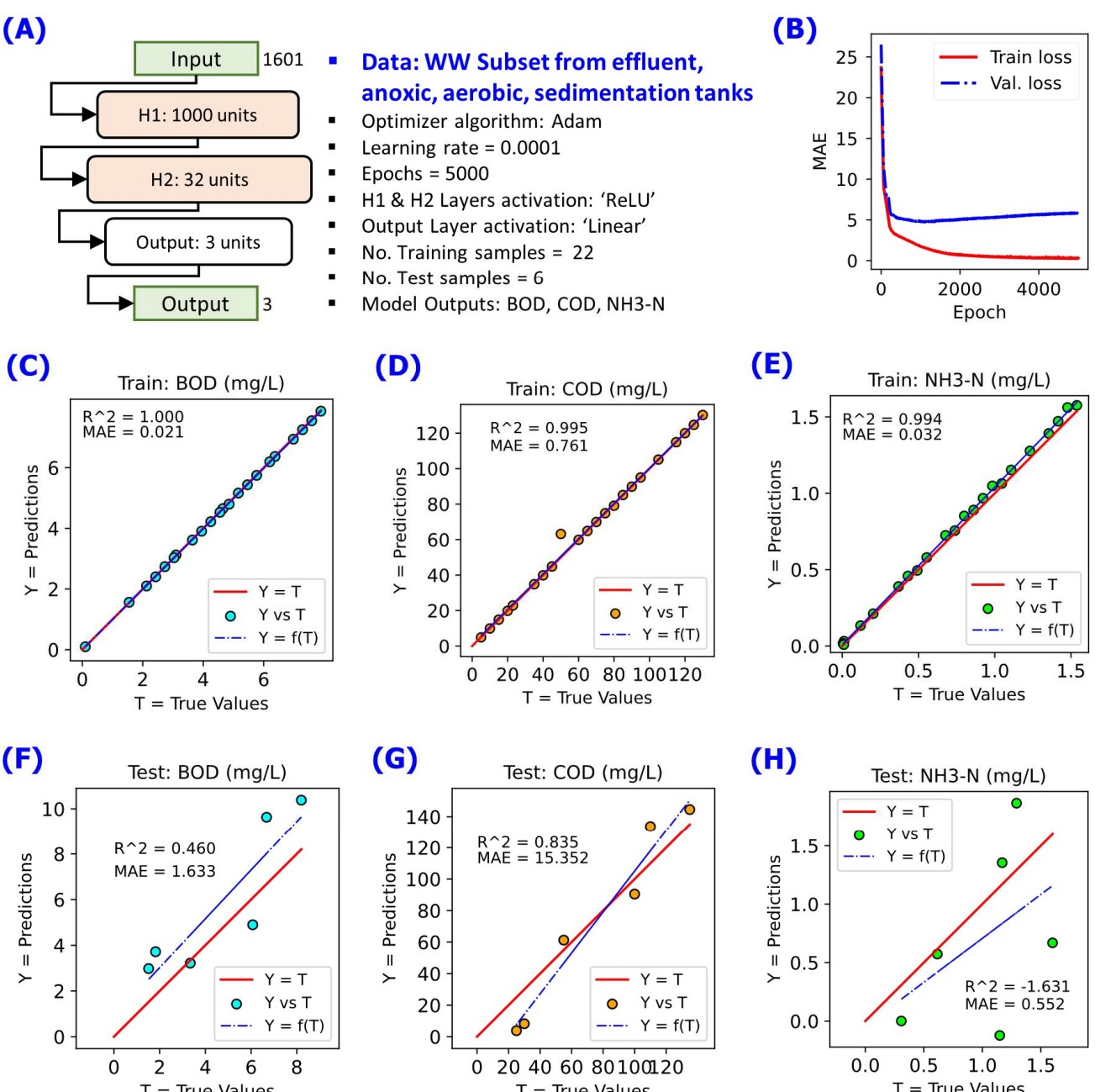

**Figure 5.** NN modelling on the data subset for Group 2 WW subset using 2 hidden layers consisting of 1000 neuron units in first hidden layer H1 and 32 neuron units in the second hidden layer H2. A linear fit Y = f(T) with a blue dashed line is annotated to compare with reference line Y = T with the red solid line. (**A**) NN model configuration implemented, (**B**) changes in cost function MAE during the training for 5000 epochs, (**C**) Y-vs-T of the training data for BOD, (**D**) Y-vs-T of the training data forID, (**E**) Y-vs-T of the training data for NH$_3$-N, (**F**) Y-vs-T of the test data for BOD, (**G**) Y-vs-T of the test data for COD, and (**H**) Y-vs-T of the test data for NH$_3$-N.

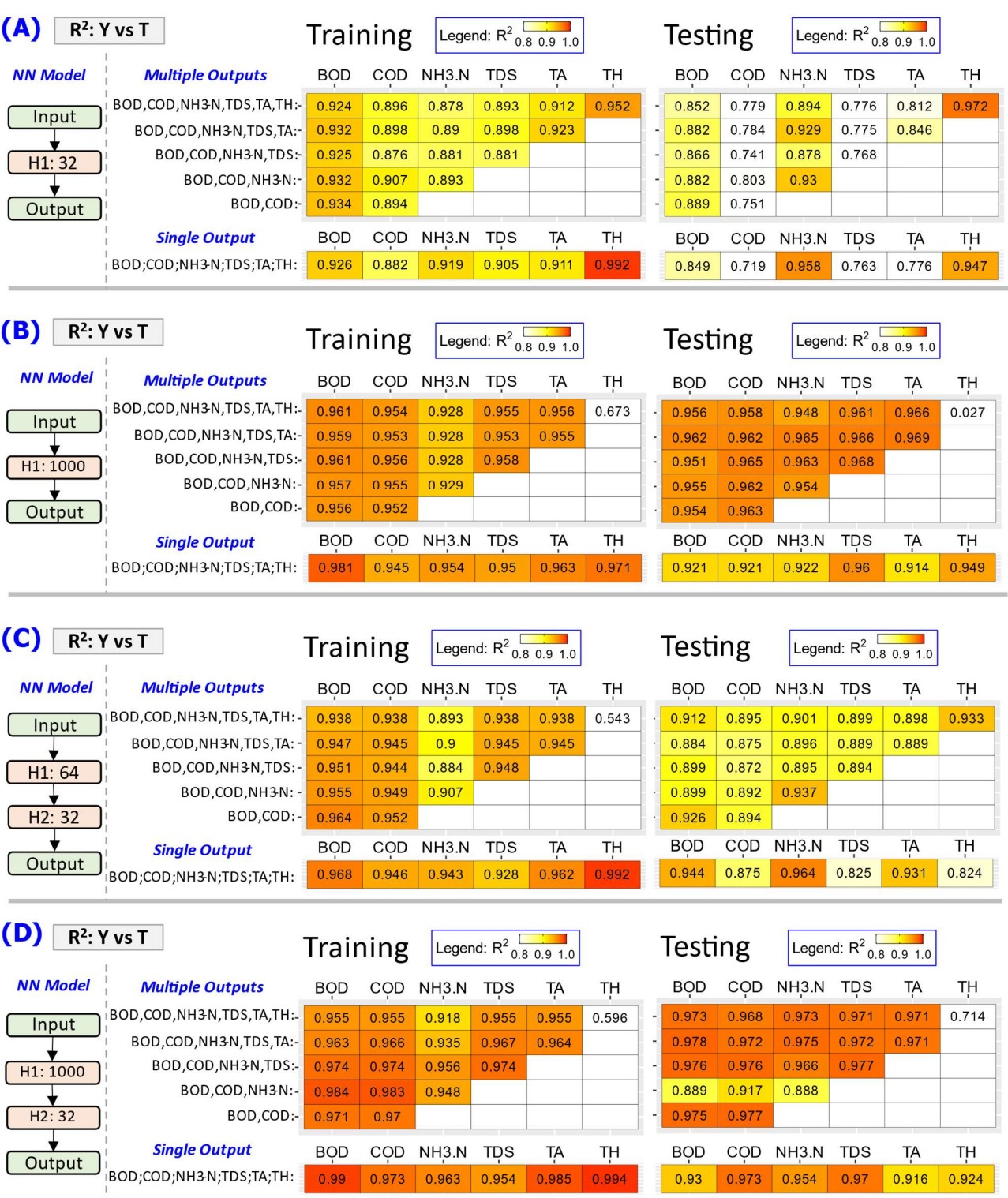

**Figure 6.** Summary of R² values between target T values and predicted Y values of the Influent WW (Group 1 WW) parameters from the training and testing of the NN model at varying model configurations and model output combinations. Color fill represents the R² values, with red being closest to 1.0 and no color fill for the R² values lower than 0.8. (**A**) Using NN model with 1 hidden layer (H1) of 32 neuron units, (**B**) using NN model with 1 hidden layer (H1) of 1000 neuron units, (**C**) using NN model with 2 hidden layers: 64 neurons units in first (H1) and 32 neuron units in second (H2), and (**D**) using NN model with 2 hidden layers: 1000 neurons units in first (H1) and 32 neuron units in second (H2).

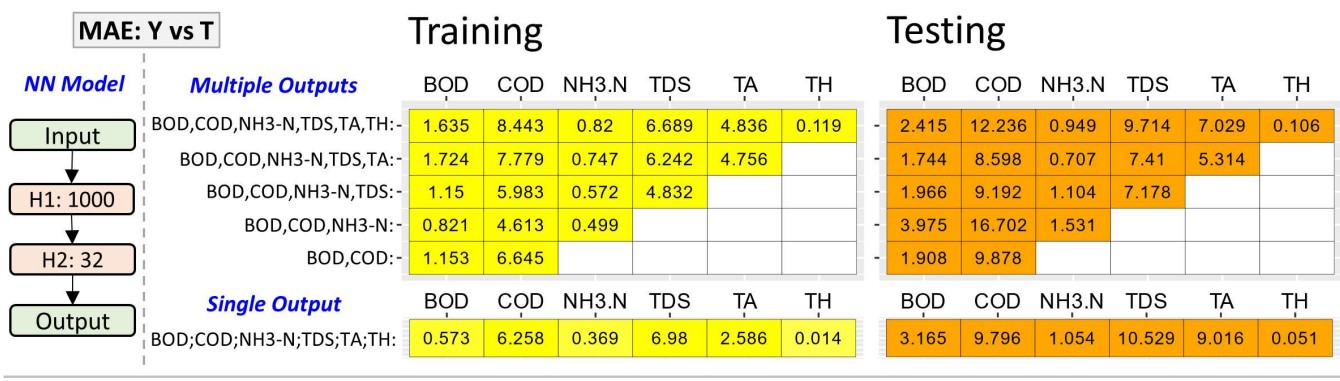

**Figure 7.** Summary of mean-absolute-error (MAE) between target T values and predicted Y values of the influent WW (Group 1 WW) parameters from the training and testing of the NN model with 2 hidden layers: 1000 neurons units in first (H1) and 32 neuron units in second (H2).

**Table 2.** Hyperparameter grid-search levels used to determine the best NN model hyperparameters for training on the Group 2 WW dataset (WW data subset for anoxic tank, aerobic tank, sedimentation tank, and WW effluent).

| NN Model Hyperparameter Grid-Search Settings | Best NN Model Hyperparameter Setting |
|---|---|
| Optimizer: ['Adam', 'Adadelta', 'SGD'] | Optimizer: 'Adam' |
| Activation function in H1: ['ReLU', 'Linear'] | Activation function in H1: 'Linear' |
| Activation function in H2: ['ReLU', 'Linear'] | Activation function in H2: 'ReLU' |
| Activation function in output layer: ['ReLU', 'Linear'] | Activation function in output layer: 'Linear' |
| Number of neuron units in H1: [1600, 1000, 64] | Number of neuron units in H1: 1000 |
| Number of neuron units in H2: [64, 32, 9] | Number of neuron units in H2: 32 |
| Learning rate *: [0.00001, 0.0001, 0.001, 0.01] | Learning rate *: 0.0001 |

Total hyperparameter grid-search settings with full-factorial grid via Scikit-Learn 'GridSearchCV' = 864
Epoch for each setting NN model training = 5000

* Note: A learning rate schedule was implemented using the tabulated value as the initial rate and the value was decreased to a factor of 0.5 (50% decrease) for every 500 epochs using an exponential decay model. See the Python codes provided for details.

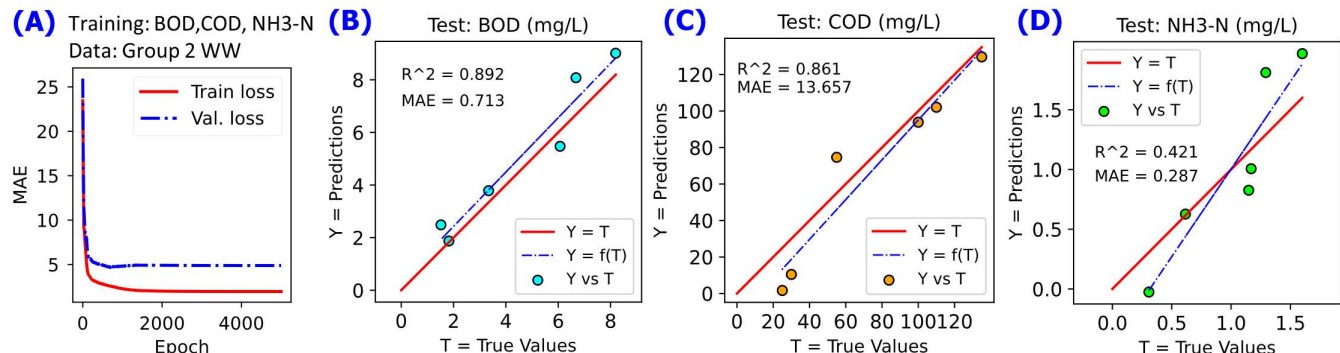

**Figure 8.** NN modelling on the data for Group 2 WW subset using the NN model configured using the best settings from the hyperparameter grid-search (see Table 2). A linear fit Y = f(T) with a blue dashed line is annotated to compare with reference line Y = T with the red solid line. (**A**) Cost function MAE value during the training for 5000 epochs, (**B**) Y-vs-T of the test data for BOD, (**C**) Y-vs-T of the test data for COD, and (**D**) Y-vs-T of the test data for NH$_3$-N.

### 3.3. Improving the NN Model via Hyperparameter Grid-Search and K-Fold Cross-Validation

Based on the results in the previous sections, the Influent WW (Group 1 WW) parameters could be modelled at good levels of fitness (high $R^2$ and low MAE) using an NN model

(Figures 6 and 7) consisting of two hidden layers with 1000 units in the first layer (H1) and 32 units in the second layer (H2). However, using the same NN model for the Group 2 WW data subset, which is the WW subset for the anoxic tank, aerobic tank, sedimentation tank and water outlet (effluent WW), did not perform as well, as can be seen in Figure 5 (for other results, see the GitHub repository of the paper [17]). A possible solution to this issue, which is also an opportunity to demonstrate the flexibility of the NN modelling approach, is the evaluation of hyperparameters beyond those used above, i.e., a learning optimizer other than Adam, a learning rate other than 0.0001, a varied number of neurons in H1 and H2, etc. The hyperparameter grid-search implemented is summarized in Table 2 with the corresponding best setting determined after the grid-search based on scoring using MAE. Figure 8 shows pertinent graphical results after training the NN model consisting of the best settings from the hyperparameter grid-search (Table 2, column 2).

When working with a small sample size is inevitable, which is the case with the Group 2 WW data subset, the performance of the NN model may also be improved by training on the whole data subset via a k-fold cross-validation approach [1]. The results are shown in Figure 9. The specific type of k-fold cross-validation implemented in this study was the repeated k-fold, which overcomes the tendency of single-pass k-fold to be noisy, especially on small datasets [27].

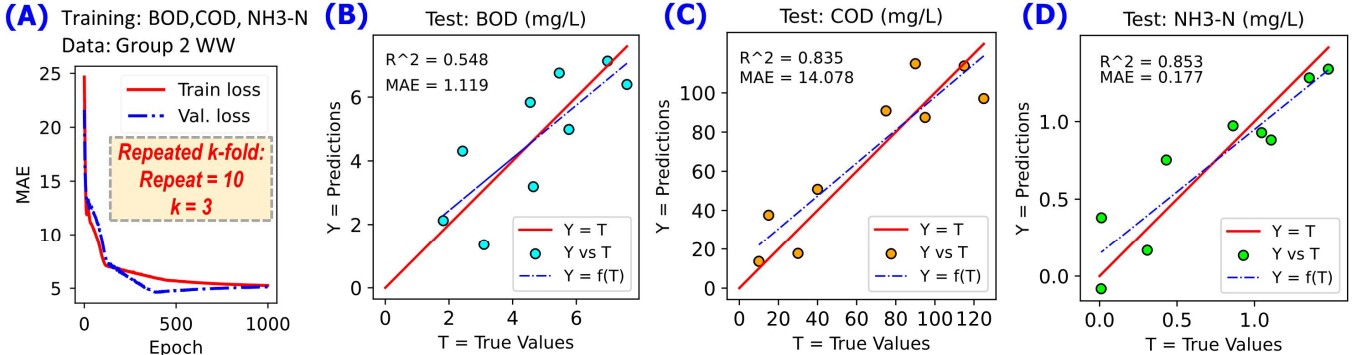

**Figure 9.** Repeated k-fold NN modelling on the data for Group 2 WW. A linear fit Y = f(T) with a blue dashed line is annotated to compare with reference line Y = T with the red solid line. (**A**) Cost function MAE value during training for 1000 epochs, (**B**) Y-vs-T of the test data for BOD, (**C**) Y-vs-T of the test data for COD, and (**D**) Y-vs-T of the test data for NH$_3$-N.

## 4. Discussion

First, we will address the main questions posed in the introduction of the paper (Section 4.1). Then, the limitations of the results of the work will be discussed (Section 4.2). Finally, we will put into perspective the significance and possible future directions of the research work (Section 4.3).

### 4.1. Answers to the Main Questions of the Study

Question 1: What data preprocessing tasks must be performed on the WW spectral data to produce regression NN models with good prediction performance?

Based on the results shown in Figures 3–5, the WW dataset of a treatment facility may need to be grouped according to WW strength levels (high strength for Group 1 WW and low strength of Group 2 WW) for a better performance of NN models. The apparent grouping of data in the plots of Y-vs-T for the whole WW dataset, as shown in Figure 3, was an interesting trend from a NN model trained for regression. Such grouping trends usually arise in the classification type of NN modelling. The grouping trends in Figure 3 indicate how the performance of a NN model can be good only when the data array being used as the input has been properly preprocessed, which, in this case, was a simple grouping of the data into subsets according to the WW strength (Group 1 WW and Group 2 WW). Another preprocessing method implemented in this study was the scaling of the banded spectral

reflectance data before feeding them as input to the NN models (see Section 2.1.2). Scaling the input features allows for each feature, which is the spectral band, in this case, to have standardized levels between 0 and 1, and this usually improves NN model training [19].

Question 2: What are the effects of some common hyperparameters in NN modeling: the number of hidden layers and the number of neuron units in each hidden layer?

The effects of the number of hidden layers and the neuron units are shown in Figures 6 and 7 for the influent WW (Group 1 WW). In general, the higher the number of neuron units in a hidden layer, the better the prediction performance. The same trend can also be seen for the effect of the number of hidden layers. This, however, does not mean that a large number of hidden layers and neuron units always results in better model performance. Other studies have shown that large numbers of hidden layers and neuron units can result in the overfitting of the model on the training set, with the consequence of poor prediction performance on the test set [28]. Therefore, the number of hidden layers and neuron units must be set accordingly, perhaps with guidance from the hyperparameter grid-search (Section 3.3).

Question 3: How does the number of modelled outputs, i.e., WW parameters, affect the prediction performance of the NN models?

In general, the NN models have their best prediction performance when the modelled output variable is a single WW parameter, i.e., BOD only, COD only, etc., as shown in Figures 6 and 7. As more WW parameters are assigned as NN model output variables, the prediction performance can degrade. The implication of this finding is that the same spectral reflectance input data can be fed to separate NN models, with each model assigned to a WW parameter to be estimated. This setup can be easily implemented in a computer running NN models. If the performance of a combination of WW parameters as outputs is acceptable, then NN models with multiple outputs may also be run for estimation.

Question 4: How can hyperparameter tuning and k-fold cross-validation on regression NN models improve prediction performance?

Hyperparameter tuning can boost the performance of a machine learning model such as an NN model [29]. The challenge with the WW data subset of Group 2 WW is apparent in Figure 5F–H, which shows that even though the NN model can have a good performance on the training set, the prediction performance of the model on the test set may be very poor. One obvious limitation in the Group 2 WW subset data is the fewer number of observations (with 28 samples) compared to the Group 1 WW subset (with 57 samples). Machine learning models such as NN models improve in their prediction capabilities with an increasing number of observations used in training. Given that this limitation on the number of observations was limited by the source of the data, hyperparameter grid-search was the only approach left to try to improve the NN model performance on the Group 2 WW subset. The results of the hyperparameter grid-search for improved prediction performance shown in Table 2 and Figure 8 indicate that there was an improvement in terms of higher $R^2$ values and lower MAE values compared to those of Figure 5F–H. Even though the Y-vs-T $R^2$ and MAE in Figure 8 for Group 2 WW were not close to those of the Group 1 WW shown in Figure 6D, the improvements compared to those in Figure 5F–H are good indications of the potential of hyperparameter grid-search to improve NN models.

Repeated k-fold cross-validation can also improve the prediction performance of the NN models, as shown in Figure 9. K-fold cross-validation can improve the reliability of a NN model, especially when the dataset size is small [30], such as that of the Group 2 WW data subset. The one drawback of k-fold cross-validation is the computational cost, because the model training is performed k times in a single-pass training [31]. This contrasts with the single training run needed to perform the single-split training in the prior tasks, with results shown in Figures 3–8.

*4.2. Limitations of the Proposed Data Analytics*

Amid the promising results of our proposed data analytics, there are limitations in what they can do. First is the inherent deficiency of the interpretability of NN models [32].

This means that NN models may not be the best models to use when the main objective of a study is the determination of the mathematical structure of the system being studied [33]. However, the interpretability of NN models should not be an issue if the objective is the process dynamics control of a WW treatment facility in which the main concern is the minimization of estimation error, e.g., MAE, of NN model prediction. If the user is more interested in determining the mathematical structure of the system variables, then non-NN modelling must be consulted, such as in these works: [7,34–36]. Another limitation is that we cannot develop a single NN model that can be applied to all the WW stream groups in a WW treatment facility, which was made apparent in the preliminary training results shown in Figure 3. Finally, the proposed data analytics being a machine-learning technique performs well when many data samples are used for training the NN models. This was made apparent in the results of NN model training for the Group 1 WW (Figure 4) and Group 2 WW (Figure 5), where the former WW subset had more data samples compared to the latter data subset. Even though hyperparameter grid-search and k-fold cross-validation (Section 3.3) may help improve model reliability and prediction performance in small sample sizes, the NN models usually improve with larger sample sizes used in training [37].

### 4.3. Perspective

Applications of machine-learning models in improving the operation of WW treatment facilities have been gaining popularity [38,39] due to the success of machine-learning models in other processing systems [40]. A bottleneck in this effort, however, is the limitation of some key process variables, e.g., BOD, COD, etc., to be measured in a timely manner such that the measurements can be used as inputs into a machine learning model for the WW treatment facility. Even a traditional non-machine-learning process control system can significantly benefit from a fast estimation of process variables [41]. This current work aimed to contribute to this need for a fast estimation technique for process variables by using spectral reflectance data of WW as the input to a data analytics workflow that uses NN regression models.

The use of spectral reflectance has been a common technique in remote sensing areas, such as in low-orbit satellites designed to study the Earth's water [36,42] and atmospheric data [43]. Though such applications are in a large-scale aggregation of signals from the sources of spectral reflectance, they inspired the concept of also applying spectral reflectance in small-scale systems such as soil systems [44,45] and water systems [7]. The processing of such spectral data to estimate target variables poses a challenge in terms of the speed of data analytics when implemented in a process control system for a WW treatment facility and similar systems that can benefit from immediate signal conversion to system variable values. Unlike the analysis performed by the originators of the dataset [7], the use of NN models in this study eliminates the intermediate steps of fitting the raw spectral data to structured models of data filters, which usually result in a loss of information from the original spectral signal and additional lag-time in the data analytics workflow. The NN models account for all signal features during the training stage.

Given a NN model has been trained, the estimation of the WW parameter levels when the spectral data are fed to the model takes just milliseconds to execute with a laptop computer of decent hardware specifications, such as the one used in this work. An envisioned practical setup can involve the use of smaller hardware that can run such computations. With the increasing adoption of artificial intelligence (AI) in various systems, hardware dedicated to running AI programs with a small physical size has been recently developed by leading companies such as NVIDIA, with their Jetson Nano for implementing AI in mobile platforms [15]. Such innovations in hardware can be leveraged to implement NN models integrated with off-the-shelf sensors for spectral reflectance in the wavelength range of 400–2000 nm used in the collection of the dataset [7] used in this study. The traditional wired sensors connected to a centralized control room where the NN model estimations are run in centralized computers can also be a default implementation setup [46]. The NN-enabled sensor system, however, will not totally eliminate the traditional WW

sample-based chemical analysis method. Such sample chemical analysis will still be carried out, but with less frequent occurrences to make sure that the NN model is within reasonable accuracy. The chemical analysis data will also be added to the training dataset where the NN model can be re-trained to refine the model parameters for better estimation.

Finally, an important consideration in the proposed approach of using NN models trained on WW spectral reflectance is the acceptable range of error on the model output for a particular application. Figure 7 shows a summary of the MAE for various NN models. The MAE values represent the average error around the true values T by the estimate Y from the NN model. These errors may be acceptable in the process control system of a WW treatment facility, but these may not be acceptable errors for purposes requiring more stringent limits of error on WW parameters, such as scheduled measurements according to regulatory agencies like US-EPA [2]. Such more stringent WW analysis should still use the established chemical analysis for the various WW parameters amid longer analysis time. Hence, there is a trade-off between tolerance for error and the speed of spectral signal conversion to WW parameter levels via NN models.

## 5. Conclusions

Neural network (NN) models may accelerate the estimation of wastewater quality parameters (BOD, COD, $NH_3$-N, TDS, TA, and TH) at minimal estimation error by using WW spectral reflectance as input data. The WW dataset in a treatment facility may need to be grouped according to WW stream strength for the best NN model training and prediction. Various model hyperparameter settings can be configured to improve the prediction performance of NN models on WW quality parameters, including the number of hidden layers, number of neurons in each hidden layer, activation functions, learning rate, and optimization algorithm. The highest $R^2$ (0.994 for training set and 0.973 for testing set) and lowest MAE (0.573 mg/L BOD, 6.258 mg/L COD, 0.369 mg/L $NH_3$-N, 6.98 mg/L TDS, 2.586 m/L TA, and 0.014 mmol/L TH) were achieved when NN models were configured for a single-output variable compared to multiple-output variables. The number of data samples may significantly affect the predictive capability of the NN regression model and more data samples will favor better prediction performance.

**Author Contributions:** Conceptualization, D.L.B.F. and W.S.; methodology, D.L.B.F. and A.T.; software, D.L.B.F. and A.T.; formal analysis, D.L.B.F., A.P.M., W.S., E.R., R.H., W.H. and M.E.Z.; data curation, D.L.B.F.; project administration, D.L.B.F.; funding acquisition, D.L.B.F., A.T. and M.E.Z. All authors have read and agreed to the published version of the manuscript.

**Funding:** This research was partially funded by the Louisiana Space Grant Consortium (LaSPACE) through the LURA sub-award PO-0000206339 under the main NASA grant number 80NSSC20M0110.

**Institutional Review Board Statement:** Not applicable.

**Informed Consent Statement:** Not applicable.

**Data Availability Statement:** The datasets and the Python codes used in this study have been made available in the online repository of the paper via GitHub [17]: https://github.com/dhanfort/WW_Spectra_NNlearning.git (accessed on 1 August 2023).

**Acknowledgments:** We thank the supportive staff and students of the Department of Chemical Engineering and the Energy Institute of Louisiana at the University of Louisiana at Lafayette. We are also grateful to LaSPACE for supporting students who pursue research in STEM projects.

**Conflicts of Interest:** The authors declare no conflict of interest. The funders had no role in the design of the study; in the collection, analyses, or interpretation of data; in the writing of the manuscript; or in the decision to publish the results.

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
