# Peer review of "Quantitating Wastewater Characteristic Parameters Using Neural Network Regression Modeling on Spectral Reflectance"

_cleantechnol, doi:10.3390/cleantechnol5040059_

Round 1
Reviewer 1 Report
The quantitating wastewater characteristic parameters using neural network regression modeling on spectral reflectance was shown in this paper. The paper demonstrates the capability of neural network regression models to estimate wastewater characteristic properties such as BOD, COD, NH3-N, TDS, TA, and TH by training on wastewater spectral reflectance in the visible to near-infrared spectrum. In this paper various neural network model configurations were evaluated in terms of regression model fitness. The mean-absolute-error was used as the metric for training and testing the neural network models, and the coefficient of determination between the model predictions and true values was also computed to measure how well the neural network models predict the true values. The work shows that the various model hyperparameter settings can be configured to improve the prediction performance of neural network models on wastewater quality parameters. In the work shows that the neural network models may accelerate the estimation of wastewater quality analysed parameters at minimal estimation error by using wastewater spectral reflectance as input data. The wastewater dataset in a treatment facility may need to be grouped according to wastewater stream strength for best neural network model training and prediction.
The research methodology was good planned.
The work is good described and the conclusions are good described.
Some suggestion follows:
- it should be good to extend the Conclusions chapter to highlight the scientific achievements of this work,
- it should be good to extend number of references.
Reviewer 2 Report
The bad quality of dataset is the most critical issue of this article, including:
1. Volume of the dataset. The open dataset used in the paper was very small, with a total of only 86 pieces of records, while the validation dataset had only 8 records. For hyperparameter validation, there are only 28 datasets and only 6 validation sets. In actual sewage treatment plants, measuring every 5 minutes by a spectral sensor will result in 288 pieces of data per sampling point per day. However, for such sparse data, the author did not use methods such as k-fold validation. The small dataset volume and inproper teatment broke the reliability of the neural network model.
2. Characteristics of the dataset. The dataset used in the article covers different sampling points in sewage treatment processes, which means it contains spatial distribution characteristics. However, in sewage treatment plant, the temporal variation characteristics are meaningful, because the instrument is usually placed in a fixed position to measure the temporal variation data. That is to say, it is necessary to analyze the characteristics of temporal data and characterize the changes of its components over time. The sample concentration and component ratio from the sensor at the same sampling point will fluctuate significantly along with the time.
3. Severe linearity of the dataset. By using seaborn. pairplot(), I draw a multivariate correlation graph in authors' shared ipynb file. It can be seen that the six variables in the dataset, including COD, BOD, and NH3-N, have strong interdependence. This is not only because the samples come from the same source, but also because there is a gradual dilution relationship in the samples. This sampling feature is completely different from the actual operation process of the sewage treatment process. The instrument in a fixed sampling point usually shows flucutate concentration and and no correlation between the time-serie samples.
4. Compare with statistic models. Neural network regression belongs to the black box model, which relies on the quality and characteristics of the dataset. Although it can predict many indicators, its interpretability is poor. On the other hand, the statistic model and kinetics based on the characteristics of chemical properties, thus it has good interpretability by few data and indicators. For the poor dataset used in this article, simple statistical methods such as PLS, should achieve satisfactory results with strong explanatory power.
In general, based on the above understanding, in the absence of reliable datasets, the above research process lacks rationality and necessity, and can only be regarded as an exercise assignment for a course.
Reviewer 3 Report
The paper presents an intriguing theme, with well-constructed arguments and compelling reported results. However, I believe that some minor revisions could enhance its overall quality and impact. I offer the following suggestions for the authors' consideration:
1- Introduction Enrichment: To provide a more comprehensive context for the study, I suggest the incorporation of additional references in the introduction section. This will help establish a stronger foundation for the research and demonstrate the existing knowledge landscape.
2- Novelty Statement Refinement: While the novelty of the research is evident, I recommend a rephrasing and expansion of the novelty statement. This will highlight the unique contributions of the study more effectively, capturing the essence of what sets it apart from existing research.
In closing, I commend the authors for their valuable work and innovative approach. These suggested improvements are intended to refine the manuscript further and elevate its impact. I am confident that with these minor changes, the paper will achieve even greater success.
The manuscript contains a few grammatical errors that should be addressed. I recommend thorough proofreading to ensure proper grammar and language usage throughout.
Reviewer 4 Report
The manuscript is well-written and very interesting. Some comments:
1. Please rewrite the abstract, including quantitative results.
2. Please discuss better concerning papers using neural networks as a tool for wastewater treatment. It is not clear the contribution and innovation of the work.
3. How about the model prediction ability? Please discuss it.
4. Also, what are the limitations of the model proposed?
5. Conclusions should be rewritten, highlighting the contribution of the work.
Round 2
Reviewer 2 Report
The authors gave the addition of k-fold to enhance the robustness of model fitting. It is a solution for dataset in limited scale, however, if consider the strong dependence of the DNN on data quality, the conclusion of the performance of the model might be incorrect and misunderstanding if there are flaws in the dataset.
The other strong issues have not been properly addressed, such as ignoring the strong linearity between the data which implied unreliable quality of the observation (although it has been published in other journals), missing control experiments using PLS as a reference model (also commercially used model in ::SCAN sensors) to support the advantages of the NN model.
Anyway, the attempts to interpret the spectra by NN is helpful for users when the dataset is good enough to achieve acceptable model in practice. So I expect the authors re-consider the quality of dataset seriously and not guarantee the data quality without quality control procedure.
BTW, I think the appendix is not necessary, MAE and R^2 is easy to introduce by simple words or references.
Reviewer 4 Report
I recommend the publication of the manuscript
Round 3
Reviewer 2 Report
The peer review of other journal and other paper can not escape the necessary procedure in this case. Without convincing the data quality, no one can trust your research and conclusion.
Author Response
Reviewer Comment:
The peer review of other journal and other paper can not escape the necessary procedure in this case. Without convincing the data quality, no one can trust your research and conclusion.
Author Response:
The use of a peer-reviewed, published and well-cited dataset (by Xing Z, et al. 2019. Quantitative estimation of wastewater quality parameters by hyperspectral band screening using GC, VIP and SPA. PeerJ 7:e8255 DOI 10.7717/peerj.8255) that did not originate from our research group should add to the reliability of the data used in this work. Collecting new dataset is outside the scope of the current work. We will not add new dataset.